# Will R&D make investors more tolerant? Analysis based on the performance forecast of Chinese listed companies

**Yixiao Chen**[1], **Yisu Wang**[2], **Huafeng Zhao**[1], **Wei Xu**[3]*

**1** University of Jinan, Jinan City, Shandong Province, China, **2** Renmin University of China, Beijing, China, **3** Nankai University, Tianjin, China

* xuwei_jn@126.com

## Abstract

In the era of innovation dividends, whether investors, as the main participants in the capital market, can tolerate enterprise innovation activities is the key to whether the capital market can help enterprises innovate. This paper takes the listed companies of Shanghai and Shenzhen A-shares in China that disclosed quantitative performance forecasts from 2016 to 2021 as samples, obtains the market reaction of performance forecasts through the event study method and uses them as proxy variables of investors' short-term performance expectations, and uses multiple regression analysis to explore the impact of corporate R&D on investors' short-term performance expectations. The results are as follows: (1) corporate R&D investment significantly reduces investors' short-term performance expectations (that is, investors have a significant tolerance effect on enterprises with higher R&D investment); (2) the increase in the shareholding ratio of institutional investors weakens the tolerance effect; and (3) with the implementation of China's innovation-driven strategy, the tolerance effect of its capital market on enterprise R&D gradually increases, especially for high-tech companies, but has a low tolerance effect on state-owned companies' R&D risk. The results show that investors in China's capital market are not completely rational in their response to corporate R&D, and investors are willing to bear more short-term performance losses for high R&D investment, which is consistent with prospect theory.

## 1. Introduction

In the era of innovation dividends, the technological innovation of enterprises and the release of innovation dividends formed by their capital accumulation are the driving forces for a country's sustained economic growth and the establishment of a competitive advantage [1, 2]. However, the risk and uncertainty of enterprise innovation activities are high and require a large amount of capital investment. Therefore, a stable and developed financial system, a sound legal system for investor protection and an effective market for intellectual property protection are needed. A large number of research results have proven that, as the most dynamic link in the economy and a platform for efficient resource allocation, the capital market plays an

**Data Availability Statement:** All relevant data are within the paper and its Supporting Information files.

**Funding:** This research was supported by The National Social Science Fund of China (16AGL008).

The funders had no role in study design, data collection and analysis, decision to publish, or preparation of the manuscript.

**Competing interests:** The authors have declared that no competing interests exist.

important role in enterprise innovation, and an effective capital market will significantly promote enterprise innovation [3]. Therefore, whether investors, as the main participants in the capital market, can treat the innovation activities of enterprises with tolerance is the key to whether the market can help the innovation activities of listed companies and is also the key to improving the innovative financing function of the capital market.

In terms of the research on the relationship between the capital market and corporate R&D, most scholars have discussed the influence of R&D on long-term market value (including long-term stock returns) [4, 5] and short-term market value [6, 7] from the perspective of causality to test whether R&D investment can increase enterprise market value. Some scholars have also observed people's attitudes toward the risk of enterprise R&D activities due to the market reaction of R&D events [8–10]. These studies have provided much empirical evidence and explanations for investors' responses to corporate R&D, but most of them are analyzed under the framework of traditional financial theories based on investors' complete rationality and risk aversion assumptions. With the continuous development of behavioral finance, an increasing number of studies show that investors' behavior is not completely rational, and investors' preference for risk is not unchanged [11, 12]. Therefore, the analysis of investors' behavior based on the hypothesis of bounded rationality will provide a new direction for the research of such problems.

However, there is still an insufficient number of studies on whether investors can sacrifice short-term returns and tolerate the shared risks of enterprise R&D. Enterprise R&D may bring about potential cash flow to the enterprise and improve its profitability and value in the future [13]. Moreover, the income from innovation investment exhibits an obvious lag [14], and it is difficult for the continuous investment of R&D resources to support the realization of the short-term income target of the enterprise. Additionally, due to information asymmetry [15–17], it is difficult for investors to know the actual situation of enterprises' R&D innovation activities. How do people choose between a certain short-term performance loss and an uncertain long-term innovation income? Daniel and Amos pointed out in prospect theory that even in the same environment, people's risk preference will be different due to their profit or loss status, and there is a "gamble" reflex effect; that is, when facing the prospect of possible loss, people have a tendency to pursue risk. When facing the prospect of profitability, people tend to avoid risks [18]. Therefore, based on prospect theory, when facing the prospect of short-term performance loss caused by enterprise R&D, investors will also tend to pursue the uncertain long-term returns brought by R&D, which will make investors more tolerant of the short-term performance of enterprises with high R&D investment. By looking for the proxy variables of investors' short-term performance expectations and analyzing the impact of corporate R&D on investors' short-term performance expectations, we can verify the tolerance effect.

The formation of China's capital market began with the establishment of the CSRC in 1992. As a typical emerging capital market, it is obviously different from the mature capital market. The most noteworthy thing is that the proportion of individual investors in China's capital market is far higher than that in the mature capital market, while the proportion of institutional investors is relatively small, which also means that investors in China's capital market are more irrational. In addition, China's economy is in a new stage of transformation from a stage of rapid growth to high-quality development, and innovation has become the key to the realization of national strategies. The government has issued a large number of policies to encourage enterprises to carry out R&D activities, which also urges investors to have a clear preference for enterprise R&D [19]. Therefore, if we explore the impact of corporate R&D on investors from the perspective of bounded rationality, China's capital market will be a very representative sample.

Since 2002, China's capital market has begun to implement a performance notice disclosure system. The compulsory disclosure of performance forecasts for listed companies on the Growth Enterprise Market (GEM) and Science and Technology Innovation Board, the compulsory disclosure of performance forecasts for listed companies on the main board with "losses", "turning losses into profits" and "performance changes of more than 50%", and the voluntary disclosure of performance forecasts for listed companies are encouraged. Disclosure is required at either the end of the fiscal year or the beginning of the following year. In contrast to an annual report, the earnings preview discloses only information related to company performance, and its release denotes the first time that the company discloses information related to annual profits, aside from the revision report. In terms of performance, information disclosure has the advantages of low noise, and market reactions are not easily distorted. Therefore, market reactions to earnings forecasts can serve as a good proxy variable for investors' short-term performance expectations.

Therefore, do investors have a tolerance effect on companies with high R&D investment in China's capital markets? Based on the above considerations, this paper takes the listed companies of Shanghai and Shenzhen A-shares in China that disclosed quantitative performance forecasts from 2016 to 2021 as samples, obtains the market reaction of the performance forecasts through the event study method to act as the proxy variable of investors' short-term performance expectations, and uses the method of multiple regression analysis to explore the impact of corporate R&D on investors' short-term performance expectations. The results show that first, investors have a significant tolerance effect on the performance of companies with high R&D investment, which reduces short-term profit expectations. These results persist even after key variable replacement, robustness tests, and endogeneity tests using propensity score matching (PSM) and the Heckman two-stage model. Second, the enhancement of investors' professionalism, that is, the increase in institutional investors' shareholding ratio, reduces this tolerance effect on R&D. Furthermore, subjective value factors, such as the formation of a national innovation environment and the recognition of innovation dividends in high-tech industries, enhance the tolerance effect of investors on the R&D of private enterprises that dare to carry out "disruptive innovation" more than they do on that of state-owned listed companies with stable innovation investment.

The main contributions of this paper are as follows. First, based on prospect theory, this study explores the relationship between corporate R&D and investors' short-term performance expectations from the perspective of bounded rationality and uses the correlation between R&D and enterprise performance to test the tolerance effect of investors in China's capital market on enterprise R&D, which enriches the relevant literature on R&D market response and expands the research perspective of the R&D market reaction. This work acts as an effective supplement to the empirical test of prospect theory in the Chinese capital market. Second, this paper uses event studies of corporate earnings forecasts as observing events. Market reaction, as a proxy variable of investors' short-term earnings expectations, is the observation value of subjective psychological variables within the behavior of observable variables, enriches the market behavior "measuring tool", and is conducive to the further study of investor behavior and preferences.

The rest of this paper is arranged as follows. The second part presents a literature review, a theoretical analysis and the research hypotheses. The third part describes the research sample selection design, data source, model design and variable definitions. The fourth part presents the main regression results, robustness test and further analysis of the moderating effect. Finally, the fifth part discusses the conclusions and research prospects.

## 2. Literature review and research hypotheses

### 2.1 Market response to R&D information disclosure

Research on market response to R&D information disclosure has been carried out mainly from the following three perspectives. The first perspective is that of empirical research, focusing on the long-term performance or stock returns of R&D investment companies, which holds that the R&D disclosed by these enterprises is positively correlated with their long-term value or stock returns [4, 5]. However, the process of investors' long-term response to R&D is complicated. On the one hand, not all evidence holds that R&D investment has a positive impact on the long-term stock returns or value of enterprises. Some empirical results show that there is no direct connection between R&D expenditure and future stock returns due to severe distortions caused by R&D cost expenses as opposed to capitalization [8]. On the other hand, because R&D investment has an obvious lag effect on average enterprise growth [5, 14], the long-term response of the market to R&D is to some extent equivalent to the response of investors after the full transmission of R&D information or even when R&D has begun to generate profits. Therefore, such research methods are mainly used to discuss the market reaction after the full transmission of R&D information.

The second perspective focuses on the impact of R&D on the short-term market value of enterprises and suggests that enterprises' innovation activities have a significant positive impact on short-term enterprise value [6, 7, 19–21]. However, some studies have shown that the market reaction to R&D disclosure varies significantly across time and regions, and in some regions or situations, R&D investment does not increase the short-term market value of enterprises [22–25]. This kind of research well discusses the market's response to enterprise R&D; however, the market value of an enterprise is the ultimate result of a large number of factors, so it is difficult to explore the specific impact of enterprise R&D on investors.

The third perspective is that of empirical research using an event research method, which focuses on the short-term stock price response to R&D events. It is found that the announcement of an R&D investment increase or product and service innovation is significantly positive in the short term [26, 27], while the announcement of the termination of an R&D project is significantly negative in the short term [17, 28]. This kind of research focuses on the market reaction of the event announcement itself, which well discusses the specific reaction of investors to the R&D announcement, but does not explain what kind of continuous impact the R&D activities of enterprises have on investors. Moreover, as most R&D information of Chinese listed companies is disclosed together with their annual reports, these reports contain much information and create considerable noise, so it is difficult to accurately measure investors' response to R&D information by using such methods [29].

In summary, the literature has performed much research on the market response of enterprise R&D, but the vast majority of the literature is based on the traditional financial theoretical framework for analysis. The assumption of the expected utility theory in the traditional financial theory about the complete rationality of investors and the type of risk preference is too strict, which has met many challenges both in economic experiments and in the real environment [12]. With the continuous development of psychology in the financial field, behavioral finance plays an increasingly important role in predicting investor behavior and explaining anomalies in capital markets. Therefore, from the perspective of bounded rationality, studying the impact of corporate R&D on investors will provide a new research direction for this issue.

### 2.2 Prospect theory

Kahneman and Tversky put forward the famous prospect theory in their article in 1979, stating that people's profit or loss state will make them react differently under the same environment.

The article also points out that, compared with taking risks to maximize profits, economic individuals are more willing to take risks to avoid losses. When profits exist, most economic individuals are risk averse; when losses occur, most economic individuals become risk takers [18]. After that, scholars expanded the theoretical analysis of the impact of prospect theory on narrow framing, probability weight function, whether profits and losses are realized [30, 31].

Regarding the application of prospect theory, many scholars have studied and explained investor behavior bias in the capital market based on prospect theory. For example, some studies introduced investor sentiment on the basis of a discrete time prospect theory model and then deduced its impact on the investor's disposal effect. They found that investor sentiment is an important factor affecting the disposal effect [11, 32]. Some studies also explored the impact of specific risk on investor preferences from the perspective of prospect theory and explained the "mystery of specific volatility" [12, 33]. In addition, some scholars have explored risk preferences in enterprise decision-making based on prospect theory and found that family enterprises are different from nonfamily enterprises in handling risk decision-making, and family enterprises tend to disperse risks when making innovative decisions [34]. However, there is no research on investors' response to corporate R&D from the perspective of prospect theory. Prospect theory provides a good theoretical framework for discussing the changes in investors' risk preferences under profit or loss prospects, but the premise is that investors' expectations can be measured. The existing research on R&D market response does not provide appropriate methods, so we need to start from the relationship between R&D and enterprise performance to find effective proxy variables.

## 2.3 R&D and enterprise performance

The influence of R&D on enterprise performance has always been a hot topic. Scholars have conducted a series of studies on this relationship. Although the internal influence process is complex and needs to be unified, a large amount of evidence shows that R&D input and output have a significant positive effect on enterprise performance [35–38]. However, studies by some scholars show that in some specific fields or regions, R&D has a negative or nonlinear impact on enterprise performance [39, 40]. Regardless of the results, all the studies point to one factor; that is, there is an obvious correlation between R&D and enterprise performance.

From the investor perspective, the correlation between R&D and enterprise performance is reflected mainly in two aspects. On the one hand, R&D affects investors' expectations of the future cash flow of enterprises and their expectations of future enterprise performance, which is directly reflected in the fact that R&D improves the market's valuation of enterprises [41–43]. On the other hand, although investors know that R&D may create additional income for enterprises, there is a serious asymmetry in R&D information between investors and enterprise management [8, 9, 44], so investors obtain information related to enterprise R&D through all possible channels to judge whether it will be successful. Therefore, in the case of information asymmetry, enterprise performance becomes a key piece of information; that is, at least in the eyes of investors, it provides relevant information about enterprise R&D to which investors can respond.

Investors' reaction to corporate performance reports is directly influenced by the performance expectations of the latter [45]. Therefore, market reaction to corporate performance reports can serve as a proxy variable of investors' performance expectations. To have an accurate measurement of investors' performance expectations, it is very important to select a good performance report as the observation event. In the Chinese market, annual reports are not good observation events because they contain a great deal of information, such as dividend plan disclosures, performance disclosures, and R&D disclosures, resulting in too much noise

in the annual reports of Chinese listed companies [29]. Some scholars have studied the market reaction to performance forecasts in China and found that companies implementing performance forecast systems have significant abnormal returns during the performance forecast disclosure period; that is, performance forecasts have significant information content [46, 47]. In addition, since the China Securities Regulatory Commission (CSRC) stipulated the embryonic form of the current performance warning system in 2002, this system has become increasingly more mature [48]. In addition, earnings forecasts disclose only performance-related information; with the advantage of low noise, market reaction to earnings forecasts are an effective proxy variable of investors' short-term performance expectations.

## 2.4 Research hypotheses

The fundamental purpose of R&D is to improve the innovation capability of enterprises so that they can gain advantages over their competitors [49]. In China's capital market, a large number of empirical studies have shown that R&D plays a significant role in promoting enterprise performance [50–52], but innovation activities are highly uncertain. Intangible asset investment has a higher probability of failure than does tangible asset investment [53]. Due to the serious asymmetry of R&D information between investors and enterprise management [8, 9, 44], it is difficult for investors to know the specific enterprise innovation situation, so the income that R&D can bring to investors is uncertain. At the same time, because a large amount of R&D investment requires a large amount of enterprise cash flow consumption, coupled with the lagged effect of R&D on enterprise performance [6] and the expense of a large amount of R&D, short-term enterprise performance is bound to be affected. According to prospect theory, Daniel and Amos pointed out that people tend to pursue risk when faced with a certain prospect of loss, which is called the reflex effect. According to the reflex effect, when facing the uncertainty of R&D and the downward pressure on the short-term performance of enterprises, investors will also tend to gamble on the long-term benefits that R&D may bring and be more tolerant of the short-term performance of enterprises [18]. In summary, this paper proposes the following hypothesis:

H1: R&D significantly reduces investors' short-term profit expectations; that is, investors have a significant tolerance effect on enterprises' innovation behaviors.

With the continuous development of the capital market, institutional investors are playing an increasingly critical role in China's capital market by virtue of their professional backgrounds and resources. At present, these investors occupy the position of the top ten shareholders in most listed companies and have a significant influence on corporate strategic decisions [54]. The influence of institutional investors on enterprise innovation is both a focus of scholars in the early stage [55] and a controversial topic [56]. "Short-sightedness theory" holds that institutional investors have a strong motivation to seek short-term returns from stock investments [57]. Poor short-term performance is likely to lead to the large-scale selling of a company's stock by institutional investors [58], thus arousing the concern of senior executives about adverse consequences such as salary reductions or even dismissal [59]. Because of these factors, when institutional investors hold shares, senior executives often drastically reduce enterprise innovation input to maintain outstanding short-term performance, which ultimately has an inhibiting effect on enterprise innovation [60]. "Sophistication or surveillance theory" holds that institutional investors can become "mature investors" by virtue of their professional advantages in terms of investment decisions, information channels and information analysis [61], and these investors pay more attention to long-term indicators (such as innovation) in making their investment decisions. Moreover, institutional investors' professional advantages in terms of information collection and interpretation enhance

enterprises' ability to supervise executives. In turn, the above situation reduces the opportunistic behavior of executives in which they avoid innovation activities, ultimately promoting enterprise innovation [62].

The tolerance effect of investors on R&D is a kind of reflex effect, which occurs only when there is certain loss and uncertain risk and a choice is made between them. Due to their professional advantages, institutional investors have relatively low information asymmetry and a more accurate judgment of enterprise innovation prospects. Therefore, compared with ordinary investors, the reflex effect of institutional investors is weakened, making them pay more attention to the short-term performance pressure brought about by R&D and more inclined to sell assets when their short-term performance is poor, which significantly weakens the tolerance effect. Therefore, this paper is more inclined toward the "short-sighted view" and proposes the following hypothesis:

H2: The shareholding ratio of institutional investors has a significant negative moderating effect on the tolerance effect.

## 3. Methodology

### 3.1 Sampling and data collection

This paper takes all A-share listed companies that disclosed their performance forecasts from 2016 to 2021 as samples. The data related to performance forecasts are from the China Stock Market & Accounting Research (CSMAR) database; patent data are from the National Bureau of Statistics; and R&D, stock price and company financial data are from the CSMAR database and TuShare big data community. The sample selection procedures are as follows: (1) samples of enterprises in the financial industry are excluded, (2) samples of enterprises with special treatment (ST/*ST) and delisting are excluded, and (3) samples with missing data are deleted. Moreover, as the focus of this paper is on the impact of innovation disclosure on market reaction to earnings forecasts, to avoid the interference of earnings forecast correction in market reactions, this paper uniformly uses the earnings forecasts released for the first time by listed companies. Due to the large fluctuation in R&D and profit forecast data, to avoid the influence of extreme values, all continuous variables are reduced by [5%, 95%]. Finally, a total of 6,363 performance forecasts are obtained.

### 3.2 Major variable construction

**3.2.1 Explained variable.** The explained variable is the cumulative excess return (*CAR*) of an enterprise within the window of the forecast event in the current year. Investors' reactions to performance reports often depend on the comparison between corporate performance changes and investors' expectations. Therefore, after controlling for the information on performance changes, the cumulative excess return (*CAR*) within the performance forecast event window can be used as an effective proxy variable for investors' short-term performance expectations. This paper uses a market model method to calculate the cumulative abnormal returns in the window period, the model of which is as follows:

$$R_{i.t} = \alpha_i + \beta_i R_{m,t} + \varepsilon_{i.t} \tag{1}$$

$$AR_{i.t} = R_{i.t} - (\hat{\alpha}_i + \hat{\beta}_i R_{m,i}) \tag{2}$$

$$CAR_i = \sum AR_{i,t} \tag{3}$$

where $R_{i.t}$ is the stock return of the company on that day and $R_{m,t}$ is the market rate of return

of the day. In this paper, the corresponding SSE Index, Shenzhen Component Index and GEM are selected to represent the market rate of return according to sample sources. $AR_{i,t}$ is the excess return rate of the company on that day. This study takes the announcement date of the performance forecast as the event date and 120 days before the event (-5, -125) as the estimation period. The ordinary least squares (OLS) method is used to estimate the parameters in model (1), and the days before and after the event (0, 3) are selected as the event window period. The abnormal returns of each day in the event window period are calculated by using model (2), while model (3) is used to calculate the cumulative abnormal returns of the whole window period.

**3.2.2 Explanatory and moderating variables.** The explanatory variable is the intensity of R&D investment (*R&D*) in the previous year. This paper focuses on the tolerance effect of Chinese capital market investors on enterprises' innovation activities. Since this tolerance effect occurs before the effect of innovation activities, it is not suitable to use proxy variables, such as patent number, to reflect innovation output. Referring to the literature, R&D investment intensity (*R&D*) is taken as a proxy variable for enterprises' innovation activities, and R&D is specifically defined as the proportion of enterprises' annual R&D expenditure in total assets in the previous year[41–43, 63].

The moderating variable is the shareholding ratio of institutional investors at the end of the year. The ratio of institutions to investors is a direct reflection of the strength of the influence of institutions on investor behavior. Referring to the literature, the ratio of institutional investors' shareholding is specifically defined as the proportion of institutional investors' shareholding in the company's circulating share capital at the end of the year [55, 56].

**3.2.3 Control variables.** To control the part of the cumulative excess earnings affected by the information on performance changes during the performance forecast event window, referring to the research of Luo and Song (2012) [47] on market response to performance forecasts, this paper uses unexpected earnings (*UE*) to measure the companies included in the performance forecast; unexpected earnings (*UE*) is used as a control variable, the calculation formula of which is as follows:

$$UE_{i,t} = (FNI_{i,t} - NI_{i,t-1})/TA_{i,t-1} \qquad (4)$$

where $FNI_{i,t}$ is the average value of the company's net profit forecast for the current year (the performance forecast discloses the profit range), $NI_{i,t-1}$ is the annual net profit of the company in the previous year, and $TA_{i,t-1}$ is the year-end total assets of the enterprise in the previous year.

In addition, referring to previous studies, the control variables also include the current ratio (*CURRENT*), return on total assets (*ROA*), financial leverage (*LEV*), and gross profit margin (*GROSS*), which are used to control the influence of corporate debt-paying ability, profitability, capital structure and operating conditions on investors' short-term profit expectations. Since performance forecasts are disclosed earlier than the annual reports in the current year, investors can observe only the annual data of the previous year at such a point, so these data are uniformly used as the control variables in this paper. In this paper, year and industry fixed effects are controlled to alleviate the endogeneity problem caused by omitted variables [19]. The specific variable definitions are shown in Table 1.

## 3.3 Multiple regression model

Because this study takes the cumulative abnormal return of the performance forecast as the proxy variable of investors' short-term performance expectations, it is necessary to control the part of the cumulative abnormal return affected by the information of performance changes.

**Table 1. Variable definitions.**

| Variable | Definition |
|---|---|
| Explained variable: | |
| $CAR_{(0,3)}$ | Cumulative abnormal returns of performance forecasts in event window period (0, 3) |
| Explanatory variables: | |
| RD | R&D expenditure in the previous fiscal year divided by the book value of total assets at the end of the previous fiscal year |
| Moderating variables: | |
| Institutional | Ratio of shares held by institutional investors to the company's outstanding share capital at the end of the year |
| Control variables: | |
| UE | (expected average net profit disclosed in the performance announcement—net profit of the previous fiscal year) to the total assets at the end of the previous fiscal year |
| CURRENT | Ratio of current assets to current liabilities at the end of the previous fiscal year |
| ROA | Ratio of net profit to total assets in the previous fiscal year |
| LEV | Ratio of total liabilities to total assets at the end of the previous fiscal year |
| GROSS | Ratio of (main business income—main business cost) to main business income in the previous fiscal year |

The traditional event study method cannot meet the needs of this study because it is difficult to study a single variable. Therefore, this study chose to use the multiple regression model. The multiple regression model used in this paper is as follows:

$$CAR_{i,(0,3)} = \beta_0 + \beta_1 UE_{i,t} + \gamma_i \sum Control + \varepsilon_{i,t} \tag{5}$$

$$CAR_{i,(0,3)} = \beta_0 + \beta_1 UE_{i,t} + \beta_2 RD_{i,t-1} + \gamma_i \sum Control + \varepsilon_{i,t} \tag{6}$$

$$CAR_{i,(0,3)} = \beta_0 + \beta_1 UE_{i,t} + \beta_2 RD_{i,t-1} + \beta_3 Institutional_{i,t} + \gamma_i \sum Control + \varepsilon_{i,t} \tag{7}$$

$$CAR_{i,(0,3)} = \beta_0 + \beta_1 UE_{i,t} + \beta_2 RD_{i,t-1} + \beta_3 Institutional_{i,t} + \beta_3 Institutional_{i,t} * RD_{i,t-1}$$
$$+ \gamma_i \sum Control + \varepsilon_{i,t} \tag{8}$$

Among the above models, model (5) is used to test the significance of cumulative abnormal returns and effectiveness of performance change information transmission within the window of performance forecast events, model (6) is used to test the influence of corporate R&D on investors' short-term performance expectations, and models (7) and (8) are used to test the moderating effect of institutional investors' shareholding ratio on the tolerance effect. To improve the accuracy of the conclusions, this work conducts industry-level clustering of standard errors [64]. If the performance information contained in the forecast is valid, then β1 in model (5) should be significantly positive; if H1 is true, then β2 in model (6) should be significantly positive; and if H2 is assumed to be true, then β3 in model (8) should be significantly negative, and the explanatory power (R2) of model (8) should be significantly higher than that of model (7).

**Table 2. Descriptive statistics of the main variables.**

| VARIABLE | N | Mean | Std. | p5 | Q1 | Median | Q3 | p95 |
|---|---|---|---|---|---|---|---|---|
| $CAR_{(0,3)}$ | 9,363 | 0.00780 | 0.213 | -0.155 | -0.0542 | -0.00732 | 0.0336 | 0.146 |
| RD | 9,363 | 0.0229 | 0.0161 | 0.00101 | 0.0110 | 0.0204 | 0.0313 | 0.0617 |
| UE | 9,363 | 0.0167 | 0.0605 | -0.104 | -0.0105 | 0.0114 | 0.0385 | 0.164 |
| CURRENT | 9,363 | 2.364 | 1.699 | 0.665 | 1.205 | 1.736 | 2.932 | 7.065 |
| GROSS | 9,363 | 29.87 | 15.90 | 6.953 | 17.62 | 27.28 | 39.08 | 65.43 |
| ROA | 9,363 | 5.950 | 6.478 | -8.024 | 2.466 | 5.446 | 9.762 | 18.91 |
| LEV | 9,363 | 39.76 | 18.61 | 10.72 | 24.29 | 38.87 | 54.10 | 73.69 |

## 4. Research findings

### 4.1 Descriptive statistics

The descriptive statistics are shown in Table 2. The forecast growth range of the average enterprise performance is 1.67% of the total assets of enterprises, and the standard deviation is 0.0605, indicating that average enterprise performance is on the rise but that its growth varies greatly. The average R&D investment of enterprises accounts for 2.29% of the total assets of enterprises, and the standard deviation is 0.0161, indicating that enterprises have significant differences in R&D investment, which is helpful for identifying the impact of different R&D investments on investors' short-term profit expectations. The mean value and standard deviation of cumulative abnormal returns of performance forecasts are 0.0078 and 0.213, respectively, indicating that the cumulative abnormal returns of different performance forecasts differ greatly.

The abnormal returns during the performance forecast event window period are shown in Fig 1. The performance forecast with positive unexpected earnings has a significantly positive excess return during the event window period, and the forecast with unexpected negative earnings is within the event window period. There is an obvious negative excess return, which preliminarily shows that the information transmission of the performance forecast is effective. Investors in the Chinese capital market have a clear reaction to the performance forecast of the listed company, and the direction of this reaction is consistent with that of unexpected profit.

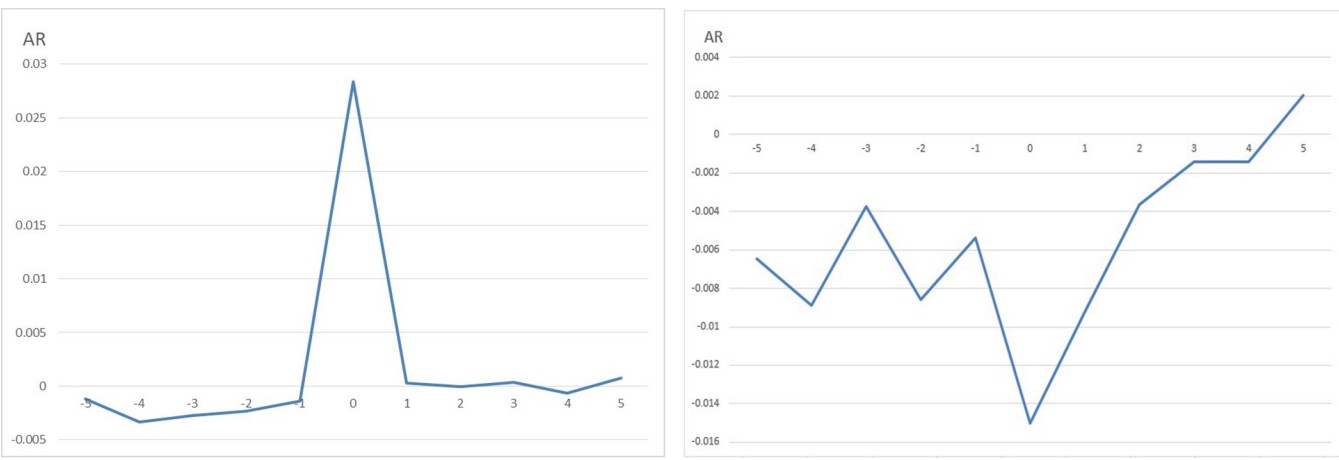

**Fig 1. (a) Abnormal returns of positive unanticipated earnings during a window period. (b) Abnormal returns of negative unanticipated earnings during a window period.**

**Table 3. Baseline regression results.**

| VARIABLE | (1) CAR$_{(0,3)}$ | (2) CAR(0,3) | (3) CAR(0,3) | (4) CAR(0,3) |
|---|---|---|---|---|
| UE | 0.049*** | 0.048*** | 0.044*** | 0.044*** |
|  | (25.01) | (24.14) | (21.60) | (21.59) |
| RD |  | 0.011*** | 0.010*** | 0.010*** |
|  |  | (9.11) | (8.86) | (9.15) |
| Institutional |  |  | 0.001 | 0.001 |
|  |  |  | (0.97) | (1.15) |
| RD*Institutional |  |  |  | -0.002*** |
|  |  |  |  | (-3.64) |
| CURRENT | -0.003** | -0.003** | -0.004*** | -0.004*** |
|  | (-2.52) | (-2.66) | (-3.16) | (-3.08) |
| GROSS | -0.005*** | -0.006*** | -0.004*** | -0.004*** |
|  | (-3.17) | (-4.69) | (-4.39) | (-4.48) |
| ROA | 0.041*** | 0.039*** | 0.034*** | 0.034*** |
|  | (25.43) | (23.40) | (20.77) | (20.98) |
| LEV | 0.002 | 0.002 | 0.002 | 0.002 |
|  | (1.05) | (0.91) | (1.26) | (1.23) |
| Constant | -0.005*** | 0.006*** | 0.007*** | 0.007*** |
|  | (-4.87) | (3.49) | (5.09) | (5.38) |
| Year F.E. | yes | yes | yes | yes |
| Industry F.E. | yes | yes | yes | yes |
| N | 9,363 | 9,363 | 9,264 | 9,264 |
| Adj_R2 | 0.0843 | 0.0864 | 0.0779 | 0.0784 |

## 4.2 Baseline results

The regression results of model (5) are shown in Column (1) of Table 3. The coefficient of UE is significantly greater than 0, indicating once again that there is a significant market reaction to performance forecasts in the Chinese market and that the direction is consistent with that of UE. The beta of the UE coefficient is estimated to be 0.017, which indicates that for every one standard deviation change in the company's disclosed unexpected earnings, the cumulative excess return during the window period changes by 1.7%. The above result is also significant in the economic sense, which means that the earnings forecast information contained is valid and that the use of earnings forecasts as observing events to measure investors' expectations for short-term performance is feasible; the appeal result is consistent with the research conclusion of Luo (2012) [47].

The regression results of model (6) are shown in Column (2) of Table 3, where the R&D coefficient is 0.011, which is positively significant at the 1% level, indicating that the cumulative abnormal returns of the performance forecast in the event window increase by 1.1% for every one-standard-deviation increase in R&D. The mean cumulative excess return of 0.78%, with 1.1% ascension, means that relative to the earnings forecast, average cumulative excess return, increasing R&D by one standard deviation, can improve cumulative abnormal returns by 141%. This result has remarkable economic meaning and verifies hypothesis H1, explaining that enterprise R&D input significantly reduces the short-term results of investor expectations. Investors have a significant tolerance effect on enterprises' innovation behavior. The appeal result is basically consistent with the conclusion of Liu (2020) [19]; that is, R&D in China's capital market will increase the short-term market value of enterprises because the reduction

in short-term performance expectations means that the short-term market value of enterprises under the same performance level will also increase accordingly.

Concerning the control variables, CURRENT and GROSS are negatively significant at the 1% level, and the stronger the ability of an enterprise to operate businesses, the lower the preliminary results of its cumulative excess return, which indicates that investors hope companies with excellent solvency and operating ability can achieve better performance in the short term. Additionally, ROA is positively significant at 1%, indicating that for enterprises with high returns on total assets, investors have lower expectations for their short-term performance, and thus, investors are more tolerant of short-term performance changes for the sake of long-term profitability.

The regression results of models (7) and (8) are shown in Columns (3) and (4) of Table 3, respectively. The adjusted R2 of model (8) is significantly higher than that of model (7), indicating that the shareholding ratio of institutional investors has a significant moderating effect. The coefficient of RD*Institutional is -0.002 and significantly negative, and that of RD*Institutional is 0.01 and significantly positive, indicating that the shareholding ratio of institutional investors has a significantly negative moderating effect; that is, enterprises with a high shareholding ratio of institutional investors still have a tolerance effect, but this effect is significantly weakened, which verifies hypothesis H2. The above results support the "myopic theory" of institutional investors, indicating that such investors in the Chinese market pay more attention to the short-term earnings of enterprises than do ordinary investors.

### 4.3 Robustness test

**4.3.1 Alternative empirical specifications.** The baseline results in Table 3 are obtained by using OLS for regression analysis. Considering that R&D is a nonnegative and continuous truncation variable, there is a zero-truncation problem [65], so a Tobit model is used for regression as part of the robustness test. In the regression process of the benchmark results, industry clustering standard errors are used. To exclude the influence of different types of standard errors on the results, robust standard errors are used for the robustness test. In the regression process of the benchmark results, all continuous variables are treated with [5%, 95%] tail reduction. To exclude the influence of different tail reduction ratios on the results, the robustness test is carried out with [1%, 99%] tail reduction ratios. The results are shown in Columns (1)-(3) of Table 4. RD is always positive and significant at the 1% level, indicating that the research results are robust for other empirical norms. The coefficient of RD regressed with the Tobit model is 0.005 higher than that of the baseline regression; that is, the cumulative excess return of the performance forecast in the event window period will increase by 0.5% every time the standard deviation of R&D increases, indicating that the tolerance effect of R&D is further improved after considering the zero-truncation problem.

**4.3.2 Alternative measures of key variables.** When the event study method is used to calculate and observe the abnormal returns of performance forecasts, it is not appropriate to specify the event window period, as doing so can easily lead to errors. The baseline results show that the event window of (0, 3) is the optimal event window. To exclude the influence of the selection of the event window period on the results, the event windows of (-1, 3) and (0, 5) are selected again for testing, the results of which are shown in columns (4)-(5), respectively, in Table 4. RD is always positively significant at the 1% level, and the coefficient is basically consistent with the baseline result, indicating that the selection of the event window period does not affect the robustness of the benchmark results.

**4.3.3 Heckman model.** Not all listed companies disclose quantitative data on their parent companies' net profit in their earnings announcements, and such natural selection tends to produce sample selection bias. To mitigate the effect of this endogeneity problem on the

**Table 4. Robustness test.**

| VARIABLE | (1) Tobit | (2) Robust | (3) Winsor(0.01) | (4) CAR(-1,3) | (5) CAR(0,5) | (6) Heckman | (7) PSM |
|---|---|---|---|---|---|---|---|
| UE | 0.079*** | 0.048*** | 0.051*** | 0.050*** | 0.051*** | 0.048*** | 0.037*** |
| | (22.62) | (11.55) | (24.57) | (24.64) | (21.07) | (23.70) | (10.73) |
| RD | 0.016*** | 0.011*** | 0.012*** | 0.011*** | 0.012*** | 0.011*** | 0.008*** |
| | (9.25) | (3.59) | (9.53) | (9.19) | (9.75) | (8.12) | (4.53) |
| CURRENT | -0.009** | -0.003 | -0.006*** | -0.003** | -0.006*** | -0.004*** | -0.004*** |
| | (-2.46) | (-1.22) | (-5.20) | (-2.62) | (-4.49) | (-3.27) | (-4.05) |
| GROSS | -0.004* | -0.006** | -0.005*** | -0.007*** | -0.005*** | -0.007*** | -0.006*** |
| | (-1.66) | (-2.50) | (-3.45) | (-5.33) | (-3.57) | (-4.97) | (-3.16) |
| ROA | 0.057*** | 0.039*** | 0.040*** | 0.041*** | 0.042*** | 0.038*** | 0.033*** |
| | (22.76) | (11.68) | (19.31) | (26.85) | (25.25) | (24.10) | (8.32) |
| LEV | 0.005 | 0.002 | 0.001 | 0.002 | 0.003 | 0.003 | 0.001 |
| | (0.95) | (0.86) | (0.34) | (1.17) | (1.33) | (1.47) | (0.38) |
| IMR | | | | | | -0.053*** | |
| | | | | | | (-5.29) | |
| Constant | -0.107*** | 0.006 | 0.002 | 0.008*** | 0.019*** | 0.025*** | 0.008** |
| | (-23.17) | (0.65) | (1.20) | (5.34) | (10.64) | (8.04) | (2.26) |
| Year F.E. | yes | yes | yes | yes | yes | yes | yes |
| Industry F.E. | yes | yes | yes | yes | yes | yes | yes |
| N | 9,363 | 9,363 | 9,363 | 9,363 | 9,363 | 9,363 | 4,828 |
| Adj(Pseudo) R2 | 0.1104. | 0.0864 | 0.0875 | 0.0883 | 0.0886 | 0.0868 | 0.0790 |

results, the Heckman two-stage method is used to modify the main model. First, a probit model is used to perform regression on the full sample containing all the performance forecasts according to model (9). Quantity is a dummy variable that takes a value of 1 when the performance forecasts disclose quantitative data on the net profit of the parent company and 0 otherwise. Control is consistent with the benchmark model. Then, the inverse Mills ratio (IMR) is calculated by predicting Quantity through the regression results. In the second-stage Heckman model, regression is performed according to model (10), the regression method and variable definitions of which are consistent with those of the baseline model.

$$Quantity = \beta_0 + \gamma_i \sum Control + \varepsilon_{i.t} \qquad (9)$$

$$CAR_{i,(0,3)} = \beta_0 + \beta_1 UE_{i,t} + \beta_2 RD_{i,t-1} + \beta_3 IMR_{i,t} + \gamma_i \sum Control + \varepsilon_{i.t} \qquad (10)$$

The second-stage regression results are shown in Column (6) of Table 4. The coefficient of the IMR is significantly negative, indicating that sample selection bias exists in performance forecasting. RD is positively significant at the 1% level, and the coefficient is basically consistent with the baseline result, indicating that the baseline results are still robust, even with the reduction in sample selection bias. In the second-stage regression, a multicollinearity test is carried out and passed, indicating that there is no multicollinearity problem in the model.

**4.3.4 PSM.** There is a one-year lag between the cumulative abnormal returns of forecast and R&D, so the endogeneity caused by the omitted variables between the two is relatively small. To further exclude the possible influence of missing variables, this study adopts PSM. Referring to the processing method of Wang [66], this paper takes the median of R&D as the benchmark and divides the samples into two subsamples, one with high R&D investment and

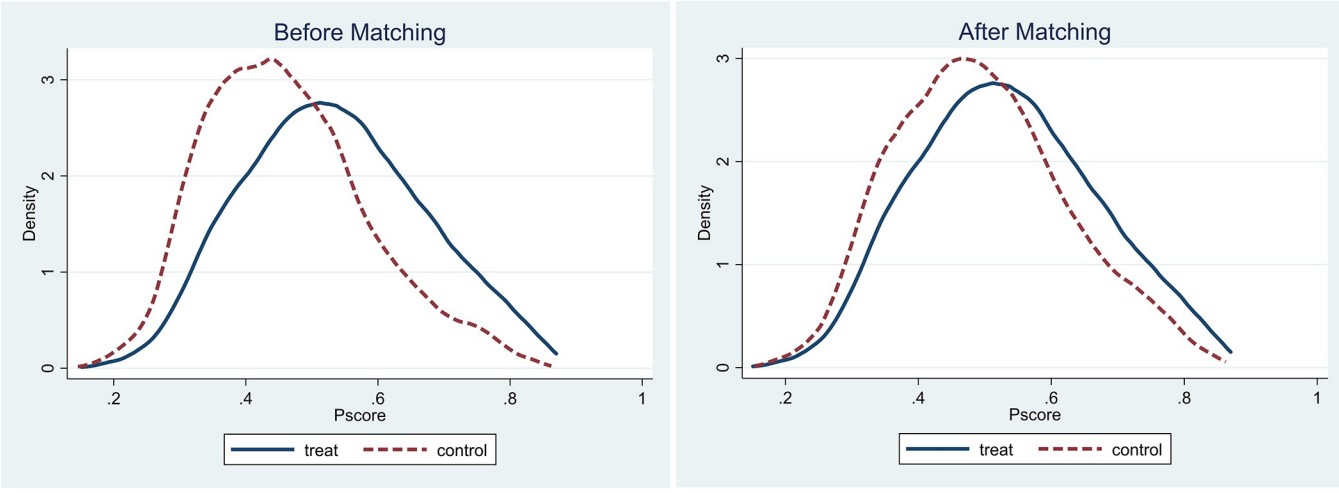

**Fig 2. Propensity score distribution before and after matching.** (a) Before matching (b) After matching.

one with low R&D investment. Enterprises with R&D higher than the benchmark are regarded as the treatment group, and those with R&D lower than the benchmark are regarded as the control group. In the PSM process, this work adopts the 1:1 nearest-neighbor matching method with putback, uses logistic regression to calculate the propensity score, and retains the set of control variables in the benchmark results.

Fig 2 shows the distribution of propensity scores before and after matching, with propensity scores on the horizontal axis and kernel density on the vertical axis. By comparing the distribution of propensity scores before and after matching, it can be intuitively found that PSM significantly corrects the bias between the treatment and control groups. Fig 3 reports the results of a comparison of such differences before and after covariate matching. The deviations of all variables are significantly reduced after matching and are all less than 5%, indicating that the balance hypothesis is satisfied. In this paper, the samples of the matched common value interval are again used for regression, and the regression model and method are consistent with the benchmark regression. The regression results are shown in Column (7) of Table 4. RD is positively significant at the 1% level, which is consistent with the baseline results; however, the coefficient decreases by 0.03 compared with the baseline regression, indicating that after considering the endogenous problem caused by missing variables, the tolerance effect is weakened, but the result is still robust.

## 4.4 Further study

**4.4.1 Impact of capital market development on the tolerance effect in China.** China's capital market has a short development period and many problems, such as a strong irrational speculative atmosphere of investors, an imperfect short-selling mechanism, serious information asymmetry and a large cognitive deviation among investors. Early studies by many scholars showed that China's stock market was invalid [67–69]. In recent years, with the continuous expansion of the overall scale of China's capital market, the continuous improvement of the level of opening to the outside world, the gradual formation and maturity of the system architecture and basic system, and changes in investors' philosophy, the short-sighted behavior of investors has improved [70], and the effectiveness of China's capital market has gradually increased [71–73]. At the same time, the innovation catch-up strategy and innovation-driven development strategy implemented by the Chinese government improve the innovation

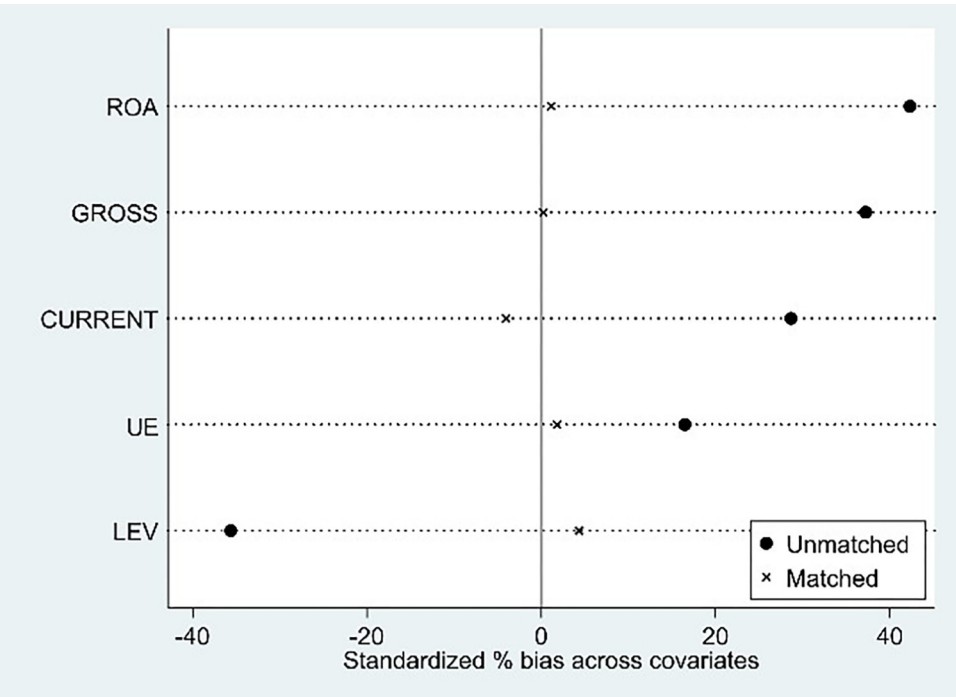

**Fig 3. Comparison of covariate differences before and after matching.**

capability of Chinese enterprises year by year [74]. The improvement of enterprises' innovation capability means that their ability to successfully carry out R&D to obtain long-term benefits is also increased, which helps improve investors' expectations of the benefits brought about by R&D. In addition, investors are more inclined toward the former considering the tradeoff between the possible long-term benefits of R&D and the downward pressure of the short-term performance of enterprises.

To further explore the change in the impact of R&D on investors' short-term performance expectations during the development of China's capital market, this work divides the samples according to year and conducts grouped regression tests. The results are shown in Table 5, where Column (1) presents the regression results of samples with years from 2016 to 2018 and Column (2) presents the regression results of samples with years from 2018 to 2021. The RD item coefficient in Column (1) is 0.005 and significantly positive, and that in Column (2) is 0.018 and significantly positive. The intergroup difference test of the RD item coefficients of the two columns is significantly positive and has significant economic significance, indicating that there is a significant tolerance effect in China's capital market from 2016 to 2021 and that this tolerance effect in the last three years has been significantly stronger than that in the first three years. That is, with the maturity of China's capital market and the formation of a national innovation environment, the tolerance effect is gradually increasing. Investors have become more focused on the long-term benefits of R&D.

**4.4.2 Effect of industry heterogeneity on tolerance.**   Often, there is a close relationship between the business scope of the enterprise and R&D intensity. There are relatively few innovative activities in many traditional industries, but the high-tech industry is knowledge intensive and technology intensive, with product diversification and rapid replacement, and enterprises must rely on new product innovation to occupy a place in a highly competitive market [75]. Therefore, investors tend to have higher expectations of long-term innovation

**Table 5. Regression results of further study.**

|  | (1) | (2) | (3) | (4) | (5) | (6) |
|---|---|---|---|---|---|---|
| VARIABLES | CAR(0,3) | CAR(0,3) | CAR(0,3) | CAR(0,3) | CAR(0,3) | CAR(0,3) |
| UE | 0.046*** | 0.049*** | 0.048*** | 0.048*** | 0.048*** | 0.048*** |
|  | (21.05) | (14.47) | (23.98) | (23.92) | (24.05) | (23.71) |
| RD | 0.005** | 0.018*** | 0.011*** | 0.007* | 0.011*** | 0.012*** |
|  | (2.53) | (4.37) | (8.52) | (2.06) | (8.98) | (7.59) |
| Tech |  |  | 0.004*** | 0.004*** |  |  |
|  |  |  | (4.08) | (3.23) |  |  |
| RD*Tech |  |  |  | 0.006* |  |  |
|  |  |  |  | (2.01) |  |  |
| SOE |  |  |  |  | -0.004*** | -0.006*** |
|  |  |  |  |  | (-3.60) | (-3.79) |
| RD*SOE |  |  |  |  |  | -0.007** |
|  |  |  |  |  |  | (-2.94) |
| Current | -0.003 | -0.002 | -0.004** | -0.004** | -0.003** | -0.003** |
|  | (-1.22) | (-0.25) | (-2.76) | (-2.91) | (-2.57) | (-2.62) |
| Gross | -0.000 | -0.010** | -0.007*** | -0.007*** | -0.006*** | -0.007*** |
|  | (-0.07) | (-2.29) | (-5.26) | (-6.03) | (-4.79) | (-4.97) |
| ROA | 0.026*** | 0.047*** | 0.040*** | 0.040*** | 0.039*** | 0.039*** |
|  | (11.45) | (11.47) | (24.29) | (25.00) | (23.31) | (22.98) |
| LEV | 0.008*** | -0.001 | 0.002 | 0.002 | 0.003 | 0.003 |
|  | (2.98) | (-0.18) | (0.97) | (0.79) | (1.28) | (1.16) |
| Constant | 0.003 | -0.002 | 0.005*** | 0.001 | 0.007*** | 0.006*** |
|  | (0.24) | (-0.06) | (3.15) | (0.37) | (4.38) | (4.57) |
| Year F.E. |  |  | yes | yes | yes | yes |
| Industry F.E. | yes | yes | yes | yes | yes | yes |
| Observations | 4,515 | 4,848 | 9,363 | 9,363 | 9,363 | 9,363 |
| Adj_R2 | 0.120 | 0.0678 | 0.0863 | 0.0869 | 0.0863 | 0.0874 |
| Suest RD(chi2) | 4.91** |  |  |  |  |  |

returns in high-tech industries, which makes them more inclined toward R&D uncertainty and the short-term performance pressure of enterprises.

To further verify the influence of industry heterogeneity on the tolerance effect, this paper refers to the industry classification standard methods of Ma Yongqiang [76] and Liu Cheng [77] and combines the information transmission, software and information technology service industry, scientific research and technology service industry, and pharmaceutical manufacturing industry. In the manufacturing industry, general equipment, special equipment, transportation equipment, electrical machinery and equipment, computer and other electronic equipment, communication equipment, instrumentation and cultural equipment and office machinery equipment are divided into high-tech industries, and their moderating effects on R&D and short-term performance expectations are tested, the regression models of which are as follows:

$$CAR_{i,(0,3)} = \beta_0 + \beta_1 UE_{i,t} + \beta_2 RD_{i,t-1} + \beta_3 Tech_{i,t} + \gamma_i \sum Control + \varepsilon_{i,t} \qquad (11)$$

$$CAR_{i,(0,3)} = \beta_0 + \beta_1 UE_{i,t} + \beta_2 RD_{i,t-1} + \beta_3 Tech_{i,t} + \beta_3 Tech_{i,t} * RD_{i,t-1} + \gamma_i \sum Control + \varepsilon_{i,t} \,(12)$$

In these models, Tech is a dummy variable of the high-tech industry. Enterprises belonging to the high-tech industry are given a value of 1, while those belonging to other industries are given a value of 0. The regression results of models (11) and (12) are shown in Columns (3) and (4) of Table 5, respectively. The adjustment R2 of model (12) is significantly higher than that of model (11), indicating that the high-tech industry exhibits a significant adjustment effect. The RD*Tech coefficient is 0.006 and significantly positive, while the R&D coefficient is 0.007 and significantly positive, indicating that the high-tech industry has a significantly positive moderating effect; that is, the tolerance effect brought about by the R&D of enterprises in the high-tech industry is significantly stronger than that of enterprises in traditional industries, and investors are more tolerant of the innovation activities of enterprises in the high-tech industry.

**4.4.3 Influence of property rights heterogeneity on the tolerance effect.**   The property attributes of different enterprises have certain differences in their innovation policies. The controlling shareholders of private enterprises, generally individuals, belong to the category of private property rights. Thus, the subject and purpose of profit distribution are specific, and shareholders prefer to obtain more benefits and a good social reputation through enterprise operation. Therefore, the motivation of such enterprises to participate in innovation is stronger [78]. In contrast, the controlling shareholder of state-owned enterprises in China is all levels of the government or its agents, and thus, the bodies involved in such profit distribution are relatively abstract. Due to "the absence of actual investor" system defects, innovation management costs are higher. In particular, significant R&D investment by state-owned enterprises needs to go through the approval process for the "Three-Importances & One-Large" system, and some state-owned enterprises experience situations of innovation efficiency loss [79]. Therefore, facing enterprises with different property rights, investors may have different choices due to R&D uncertainty and short-term performance pressure.

To further explore the impact of property rights heterogeneity on the tolerance effect, this paper examines the moderating effect of property rights attributes on R&D and short-term performance expectations. The regression model is as follows:

$$CAR_{i,(0,3)} = \beta_0 + \beta_1 UE_{i,t} + \beta_2 RD_{i,t-1} + \beta_3 SOE_{i,t} + \gamma_i \sum Control + \varepsilon_{i,t} \tag{13}$$

$$CAR_{i,(0,3)} = \beta_0 + \beta_1 UE_{i,t} + \beta_2 RD_{i,t-1} + \beta_3 SOE_{i,t} + \beta_3 SOE_{i,t}*RD_{i,t-1} + \gamma_i \sum Control + \varepsilon_{i,t} \tag{14}$$

SOE is the dummy variable for property rights, which is equal to 1 for state-owned enterprises and 0 for non-state-owned enterprises. The regression results of models (13) and (14) are shown in Columns (5) and (6) of Table 5, respectively. The adjustment R2 of model (14) is significantly higher than that of model (13), indicating that property rights attributes have a significant moderating effect. In model (14), the coefficient of RD*SOE is 0.007 and significantly negative, and that of RD is 0.012 and significantly positive, indicating that property rights attributes have a significant negative moderating effect; that is, the tolerance effect brought about by the R&D of state-owned enterprises is obviously weaker than that of private enterprises, and investors are more tolerant of the innovation activities of the latter.

## 5 Conclusions and discussion

### 5.1 Conclusions

As an important platform through which to improve resource allocation efficiency, the capital market plays a key role in assisting enterprise R&D and innovation and driving economic growth. It is very important to explore the attitude of investors toward enterprise R&D to

understand their behavior and preferences. This work takes Chinese listed companies that disclosed quantitative earnings forecasts from 2016 to 2021 as samples, uses an event study method and observes market reactions to earnings forecasts to obtain the proxy variable of investors' short-term earnings expectations, and explores the impact of corporate R&D on investors' short-term earnings expectations. The results show that the information contained in China's capital market performance forecast is effective, which supports the research conclusion of Luo (2012) [47]. R&D significantly reduces the short-term profit expectations of investors; that is, investors exhibit a significant tolerance effect on R&D. This finding shows that investors in China's capital market are willing to take short-term risks for the long-term benefits that may be brought about by R&D. This conclusion is basically consistent with the research findings of Liu (2020) [19]; that is, R&D in China's capital market will improve the short-term market value of enterprises because the reduction of short-term performance expectations means that the short-term market value of enterprises at the same performance level will also increase accordingly. This also shows that the impact of R&D on the short-term market value of enterprises is partly caused by the irrationality of investors.

At the same time, this study finds that due to the relatively weak reflex effect of institutional investors, the shareholding ratio of institutional investors has a significantly negative moderating effect, and the tolerance effect of enterprises with a high shareholding ratio of institutional investors is significantly weakened, which supports the "short-sighted theory" of institutional investors. In this paper, the possible endogeneity problems are tested by using PSM and the Heckman two-stage model. The robustness of the results is tested by substituting key variables and empirical norms. The results are consistent, indicating that the conclusions of this study are robust. On this basis, this study further discusses the impact of capital market development, industry heterogeneity and property rights heterogeneity on tolerance. It is found that with the development of China's capital market, the tolerance effect of investors on enterprise R&D is gradually increasing. The high-tech industry has a significant positive moderating effect, and investors have a more obvious tolerance effect on the R&D of enterprises in the high-tech industry than on that of enterprises in other industries. State-owned property rights attributes have a significantly negative moderating effect, and investors have a more obvious tolerance effect on private enterprises' R&D than on that of other types of enterprises.

## 5.2 Discussion

First, the results of this study show that investors in China's capital market are not completely rational in their response to corporate R&D, and investors are willing to bear more short-term performance losses for high R&D investment, which means that prospect theory is also applicable to China's capital market. According to the influence of institutional investors' shareholding ratio on the tolerance effect, it is not difficult to find that the tolerance effect is affected by investors' rationality. If investors' overall rationality is low, the tolerance effect is high. The higher the rationality is, the lower the tolerance effect is. At the same time, the results of further research in this paper show that the tolerance effect is also affected by investors' preference for R&D, so private enterprises and high-tech industries will have a more obvious tolerance effect. Based on the above conclusions, the tolerance effect will be more obvious in the capital market, where investors are less rational but have a clear preference for R&D, similar to the Chinese capital market. Therefore, the findings of this study are more applicable to emerging capital markets with more adequate innovation guidance and investment. For mature capital markets and emerging capital markets with insufficient innovation guidance and investment, the tolerance effect may not be significant.

The research conclusion of this paper has important theoretical and practical significance. First, this paper enriches the relevant literature on R&D market reactions. Different from the

previous literature from the perspective of the complete rationality of traditional finance, this paper, from the perspective of bounded rationality, uses the correlation between R&D and corporate performance to obtain proxy variables of investors' short-term performance expectations, explores the impact of corporate R&D on investors' short-term performance expectations, and finally verifies the existence of a tolerance effect in China's capital market. In a practical sense, this study combines the two most important fields—enterprise innovation and the capital market—in the development of China's market economy, provides empirical support for the innovative financing functions of the capital market, and affirms the positive role played by China's capital market in facilitating enterprise innovation.

The biggest limitation of this study is that it only focuses on the impact of a single indicator, enterprise R&D investment, on investors' short-term earnings expectations. However, there are many types of innovative activities of enterprises and various innovative information disclosed. It is difficult to say whether different R&D information will have the same impact on investors. Therefore, in future research, we can further explore whether different types of enterprise innovation activities (such as patent output and new projects), as well as enterprise R&D efficiency and flexibility, will have tolerance effects. In addition, this study provides an effective proxy variable for investors' short-term performance expectations. Therefore, in future work, we cannot be limited to exploring the impact of innovation on investors, such as corporate governance and social responsibility, which can become potential research directions.

## Supporting information

**S1 Data.**
(XLSX)

## Author Contributions

**Conceptualization:** Yixiao Chen.

**Data curation:** Yixiao Chen.

**Formal analysis:** Yixiao Chen.

**Funding acquisition:** Wei Xu.

**Methodology:** Wei Xu.

**Project administration:** Wei Xu.

**Resources:** Wei Xu.

**Supervision:** Wei Xu.

**Validation:** Yisu Wang.

**Writing – original draft:** Yixiao Chen.

**Writing – review & editing:** Yisu Wang, Huafeng Zhao, Wei Xu.

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
