## [Decision Letter · Decision Letter 0]

22 Sep 2022

PONE-D-22-20174Will R&D make investors more tolerant? Analysis based on the performance forecast of Chinese listed companiesPLOS ONE

Dear Dr. Xu,

Thank you for submitting your manuscript to PLOS ONE. After careful consideration, we feel that it has merit but does not fully meet PLOS ONE’s publication criteria as it currently stands. Therefore, we invite you to submit a revised version of the manuscript that addresses the points raised during the review process.

We look forward to receiving your revised manuscript.

Kind regards,

Maurizio Fiaschetti

Academic Editor

PLOS ONE

Journal Requirements:

Additional Editor Comments:

The reviewers have raised some good point, therefore I would encourage you to address all of them thoroughly.

Reviewers' comments:

Reviewer's Responses to Questions

**Comments to the Author**

1. Is the manuscript technically sound, and do the data support the conclusions?

Reviewer #1: Yes

Reviewer #2: Yes

2. Has the statistical analysis been performed appropriately and rigorously? 

Reviewer #1: Yes

Reviewer #2: I Don't Know

3. Have the authors made all data underlying the findings in their manuscript fully available?

Reviewer #1: Yes

Reviewer #2: Yes

4. Is the manuscript presented in an intelligible fashion and written in standard English?

Reviewer #1: No

Reviewer #2: Yes

5. Review Comments to the Author

Reviewer #1: This paper explores the impact of corporate R&D on investors' short-term performance expectations by analyzing listed companies that disclosed quantitative performance forecasts from 2016 to 2021 in the Shanghai and Shenzhen A-share markets. The results suggest that corporate R&D investment significantly reduces investors' short-term performance expectations that is, investors have a significant tolerance effect on enterprises with higher R&D investment.

I believe that the paper needs improve argument development, formal alterations and more work to be done before publication in this reputed journal.

Major Points:

• It is not clear what is the problem at hand. How this work differs from the previous work done in this area? It needs sharp contextualization.

• This paper takes all A-share listed companies that disclosed their performance forecasts from 2016 to 2021 as samples. Contextualization from the context of sample firms and the country is missing. Why do authors rely on Chinese firms only? The authors should add a paragraph on the dynamics and importance of focusing on this sample.

• Make the literature section comparable that how your work is superior to previous work. Also, no theoretical underpinnings are provided in the literature section and empirical literature included in the paper does not cover recent evidence.

• The study applied a multiple regression model to analyse the data. How it is well suited for achieving the study’s objectives. The authors should add a significant justification for why you picked this estimation model.

• It is good that the authors have employed robustness analysis with different models and measurements of variables. However, the economic significance and discussion of the main results as well as of the robustness analysis is missing. The findings and discussion section does not provide the needed explanation and support from previous studies. The key findings should be compared with the existing studies, and they should highlight whether these findings are consistent or the other way around.

• Also, I encourage authors to develop the practical implications specifically based on the findings of the study instead of general discussion. Authors need to be more specific when it comes to possible future extensions of their work.

• There are grammatical mistakes in the paper. Please carefully go through the paper for in-text citation errors, spelling and grammatical mistakes before submitting the revised version.

Reviewer #2: The topic of the manuscript is interesting. Studying the behavior of investors and the factors that influence this behavior is important in terms of identifying barriers to economic development. However, the manuscript could be improved, my comments are attached:

- it is necessary to clarify the research methods used in the abstract, also describe the general conclusion based on the results (in the context of significance)

- it is recommended to revise the structure of the Introduction, focusing on a consistent description of the relevance, scientific and practical problems, bright points of view on this problem (other studies) with a transition to the purpose of the study and tasks, study design. The authors also need to clarify the choice of China as a field for empirical research (in terms of relevance and the possibility of applying the results to other countries)

- section 2.3 Research hypotheses is closer to 3. Methodology than to 2. Literature review

- authors are advised to separate the Discussion section and the Conclusions section, therefore section 5 needs to be revised

- a clear clarification of research limitations and assumptions is required (including making a reasonable conclusion about the possibility of extrapolating the results for other countries, and not just for China)

- the data file could be improved for reader accessibility (use English). It is recommended to add notes for headings with symbols; also add sheet descriptions and sheet titles.

6. PLOS authors have the option to publish the peer review history of their article (what does this mean?). If published, this will include your full peer review and any attached files.

Reviewer #1: **Yes: **Prof. Ahmed Imran Hunjra

Reviewer #2: No

---

## [Author Response · Author response to Decision Letter 0]

6 Nov 2022

Dear Prof. Fiaschetti and Reviewers:

Thank you for your letter and the reviewers’ comments concerning our manuscript. These comments are valuable and very helpful. We have read through the comments carefully and have made corrections. Based on the instructions provided in your letter, we uploaded the file of the revised manuscript. Revisions in the text are shown using yellow highlighting, and the responses to the reviewer's comments are presented as follows.

We are grateful to you for allowing us to resubmit a revised copy of the manuscript, and we highly appreciate your time and consideration.

Sincerely.

Wei Xu.

Response to the reviewer's comments:

Reviewer #1:

This paper explores the impact of corporate R&D on investors' short-term performance expectations by analyzing listed companies that disclosed quantitative performance forecasts from 2016 to 2021 in the Shanghai and Shenzhen A-share markets. The results suggest that corporate R&D investment significantly reduces investors' short-term performance expectations that is, investors have a significant tolerance effect on enterprises with higher R&D investment.

Response: Thank you very much for the positive comments and constructive suggestions. Please find the following detailed responses to your comments and suggestions.

Q1. It is not clear what is the problem at hand. How this work differs from the previous work done in this area? It needs sharp contextualization.

Response: Thank you very much for your comment, which is highly appreciated. The previous research on the market reaction of R&D is mostly based on the assumption of the complete rationality of investors and risk aversion under the framework of traditional financial theory. Our work analyzes this problem based on prospect theory under the assumption of limited rationality of investors, which is a new research perspective. This point was not well discussed in the previous manuscript, so we revised the relevant parts in the introduction and literature review.

Q2. This paper takes all A-share listed companies that disclosed their performance forecasts from 2016 to 2021 as samples. Contextualization from the context of sample firms and the country is missing. Why do authors rely on Chinese firms only? The authors should add a paragraph on the dynamics and importance of focusing on this sample.

Response: Thank you for your constructive suggestion. As a typical emerging capital market, China's capital market is characterized by a high proportion of individual investors, which also means that the overall rationality of investors is relatively low. At the same time, with the development of innovation in China, the guidance and investment of China's capital market for innovation are also very obvious. Therefore, if we explore the impact of corporate R&D on investors from the perspective of bounded rationality, China's capital market will be a very representative sample. In the introduction, we added a background introduction and corresponding discussion and discussed the effectiveness and limitations of using this sample in the discussion section.

Q3. Make the literature section comparable that how your work is superior to previous work. Also, no theoretical underpinnings are provided in the literature section and empirical literature included in the paper does not cover recent evidence.

Response: Thank you for your kind suggestions. The question of how our work is superior to previous work has been answered in question 1, so it will not be repeated. We have added a literature review on prospect theory and adjusted the logic of the overall literature review. In addition, we have supplemented the recent empirical literature.

Q4.The study applied a multiple regression model to analyse the data. How it is well suited for achieving the study’s objectives. The authors should add a significant justification for why you picked this estimation model.

Response: Thank you for your kind question. In this study, the cumulative abnormal return of the performance forecast is used as the proxy variable of investors' short-term performance expectations, so the part of the cumulative abnormal return affected by the performance change information must be controlled. The traditional event study method is that the study of a single variable has difficulty meeting the needs of this study, so we choose to use a multiple regression model. We have added a corresponding discussion to the multiple regression model.

Q5.It is good that the authors have employed robustness analysis with different models and measurements of variables. However, the economic significance and discussion of the main results as well as of the robustness analysis is missing. The findings and discussion section does not provide the needed explanation and support from previous studies. The key findings should be compared with the existing studies, and they should highlight whether these findings are consistent or the other way around.

Response: Thank you for your kind suggestions. In the research results section, we added a discussion of the economic significance of the empirical results (including the robustness test), and in the research results and conclusions sections, we added a comparison between the key research findings and existing research results.

Q6.Also, I encourage authors to develop the practical implications specifically based on the findings of the study instead of general discussion. Authors need to be more specific when it comes to possible future extensions of their work.

Response: Thank you for your constructive suggestion. We divided the conclusion and discussion into two sections, and in the discussion section, we provided a deeper discussion of the research results of this paper (including the conditions for the generation of the tolerance effect, as well as the applicability and limitations of the results). In addition, we have also revised the part related to future work to make it more specific.

Q7.There are grammatical mistakes in the paper. Please carefully go through the paper for in-text citation errors, spelling and grammatical mistakes before submitting the revised version.

Response: We are very sorry for our incorrect writing. To address these issues, we found a professional organization to edit the manuscript.

Reviewer #2:

The topic of the manuscript is interesting. Studying the behavior of investors and the factors that influence this behavior is important in terms of identifying barriers to economic development. However, the manuscript could be improved, my comments are attached:

Q1. it is necessary to clarify the research methods used in the abstract, also describe the general conclusion based on the results (in the context of significance)

Response: Thank you very much for your constructive suggestion. The abstract is truly important for a paper. We added general conclusions to the abstract and described the research methods in more detail.

Q2.it is recommended to revise the structure of the Introduction, focusing on a consistent description of the relevance, scientific and practical problems, bright points of view on this problem (other studies) with a transition to the purpose of the study and tasks, study design. The authors also need to clarify the choice of China as a field for empirical research (in terms of relevance and the possibility of applying the results to other countries)

Response: Thank you for your kind suggestions, which are highly appreciated. We made a comprehensive adjustment to the structure of the introduction to better lead to the purpose and innovation of this study. In addition, in the introduction, we also added the background of the Chinese sample and the reasons for its selection, and in the discussion, we added an analysis of the applicability and limitations of the results.

Q3. section 2.3 Research hypotheses is closer to 3. Methodology than to 2. Literature review

Response: Thank you for your kind suggestions. We referred to some other papers of the same type published on PLOS One and found that many of them put the literature review and research hypothesis in the same section. Therefore, we retained the original framework and changed the title of the second part to “Literature review and research hypotheses”.

Q4.authors are advised to separate the Discussion section and the Conclusions section, therefore section 5 needs to be revised

Response: Thank you for your constructive suggestion. We split section 5 according to the suggestions and performed a more profound analysis of the research results in the discussion section.

Q5. a clear clarification of research limitations and assumptions is required (including making a reasonable conclusion about the possibility of extrapolating the results for other countries, and not just for China)

Response: Thank you for your kind suggestion. In the discussion section, we added an analysis of the applicability and limitations of the research conclusions and discussed the possibility of applying the tolerance effect to the capital markets of other countries.

Q6.the data file could be improved for reader accessibility (use English). It is recommended to add notes for headings with symbols; also add sheet descriptions and sheet titles.

Response: We apologize for our negligence in the accessibility of our data. We reorganized the data files, deleted unnecessary sheets, and used all English. Unclear variable names have been modified to make them consistent with the paper. We also added a title to the chart.

---

## [Decision Letter · Decision Letter 1]

4 Dec 2022

PONE-D-22-20174R1Will R&D make investors more tolerant? Analysis based on the performance forecast of Chinese listed companiesPLOS ONE

Dear Dr. Xu,

Thank you for submitting your manuscript to PLOS ONE. After careful consideration, we feel that it has merit but does not fully meet PLOS ONE’s publication criteria as it currently stands. Therefore, we invite you to submit a revised version of the manuscript that addresses the points raised during the review process.

We look forward to receiving your revised manuscript.

Kind regards,

Maurizio Fiaschetti

Academic Editor

PLOS ONE

Journal Requirements:

Additional Editor Comments:

Your paper is well strucutred and written and it adds a valuable contribution to the debate. Reviewer 2 has some minor issues I would strongly encourage you to quickly address before resubmitting your work.

Reviewers' comments:

Reviewer's Responses to Questions

**Comments to the Author**

1. If the authors have adequately addressed your comments raised in a previous round of review and you feel that this manuscript is now acceptable for publication, you may indicate that here to bypass the “Comments to the Author” section, enter your conflict of interest statement in the “Confidential to Editor” section, and submit your "Accept" recommendation.

Reviewer #2: All comments have been addressed

Reviewer #3: All comments have been addressed

2. Is the manuscript technically sound, and do the data support the conclusions?

Reviewer #2: Yes

Reviewer #3: Yes

3. Has the statistical analysis been performed appropriately and rigorously? 

Reviewer #2: I Don't Know

Reviewer #3: Yes

4. Have the authors made all data underlying the findings in their manuscript fully available?

Reviewer #2: Yes

Reviewer #3: Yes

5. Is the manuscript presented in an intelligible fashion and written in standard English?

Reviewer #2: Yes

Reviewer #3: Yes

6. Review Comments to the Author

Reviewer #2: The authors did a great job and provided a revised manuscript, taking into account the comments of the reviewers. However, there are a few items that could be improved (minor revision):

- In Section 2.3, the authors gave a description of the H1 hypothesis, but did not provide a description of the H2 hypothesis.

- Authors need to provide links to sources of primary data. In section 3.1, the authors provide data resources, but no references.

- Authors are advised to clarify the sample size. "This paper takes all A-share listed companies that disclosed their performance forecasts from 2016 to 2021 as samples." How many such companies?

- Since a sample has some specific characteristics that determine its quality, researchers may be interested in the question of the coverage of the dissemination of the results. The authors are advised to briefly clarify the representativeness of the results (through reliability indicators, calculation of sampling error, etc., if applicable in this case).

Reviewer #3: This is an interesting paper. All suggestions are carefully fixed and I recommend to publish this interesting paper.

7. PLOS authors have the option to publish the peer review history of their article (what does this mean?). If published, this will include your full peer review and any attached files.

Reviewer #2: No

Reviewer #3: No

---

## [Author Response · Author response to Decision Letter 1]

15 Dec 2022

Dear Prof. Fiaschetti and reviewers:

Thank you for your letter and the reviewers’ comments concerning our manuscript. These new comments were very helpful in making the paper more rigorous. We have read through the comments carefully and have made corrections. Based on the instructions provided in your letter, we uploaded the file of the revised manuscript. The revisions made to the text are highlighted in yellow, and the responses to the reviewers' comments are presented as follows.

We are grateful that you have allowed us to resubmit a revised copy of the manuscript, and we greatly appreciate your time and consideration.

Sincerely,

Wei Xu

Response to the reviewers' comments:

Reviewer #2:

The authors did a great job and provided a revised manuscript, taking into account the comments of the reviewers. However, there are a few items that could be improved (minor revision):

Response: Thank you very much for the positive comments and constructive suggestions. Please find the following detailed responses to your comments and suggestions.

Q1. In Section 2.3, the authors gave a description of the H1 hypothesis, but did not provide a description of the H2 hypothesis.

Response: Thank you very much for your comment; this section was not rigorous enough. Therefore, we added a description of H2.

Q2. Authors need to provide links to sources of primary data. In section 3.1, the authors provide data resources, but no references.

Response: Thank you for your constructive suggestion. We added links to our sources of primary data in section 3.1.

Q3. Authors are advised to clarify the sample size. "This paper takes all A-share listed companies that disclosed their performance forecasts from 2016 to 2021 as samples." How many such companies?

Response: Thank you for your kind question. When we checked this section, we found that there was a problem with the description of the sample size (we forgot to modify it when updating the sample). Therefore, we corrected the error and described the sample size in more detail.

Q4: Since a sample has some specific characteristics that determine its quality, researchers may be interested in the question of the coverage of the dissemination of the results. The authors are advised to briefly clarify the representativeness of the results (through reliability indicators, calculation of sampling error, etc., if applicable in this case).

Response: Thank you for your kind suggestions. We also believe that the quality of samples has a decisive impact on the reliability and representativeness of research results. Therefore, we used the Heckman model in section 4.3.3 to reduce the impact of sample selection bias on the research results. The regression results of the Heckman model show that selection bias has no significant impact on our sample quality. To explain this point more clearly, we added a description in section 4.3.3.

Reviewer #3:

This is an interesting paper. All suggestions are carefully fixed and I recommend to publish this interesting paper.

Response: Thank you very much for your recognition of our work.

---

## [Decision Letter · Decision Letter 2]

26 Dec 2022

Will R&D make investors more tolerant? Analysis based on the performance forecast of Chinese listed companies

PONE-D-22-20174R2

Dear Dr. Xu,

We’re pleased to inform you that your manuscript has been judged scientifically suitable for publication and will be formally accepted for publication once it meets all outstanding technical requirements and you fix those minor issues flagged up by two reviewers.

Kind regards,

Maurizio Fiaschetti

Academic Editor

PLOS ONE

Additional Editor Comments (optional):

Reviewers' comments:

Reviewer's Responses to Questions

**Comments to the Author**

1. If the authors have adequately addressed your comments raised in a previous round of review and you feel that this manuscript is now acceptable for publication, you may indicate that here to bypass the “Comments to the Author” section, enter your conflict of interest statement in the “Confidential to Editor” section, and submit your "Accept" recommendation.

Reviewer #2: All comments have been addressed

Reviewer #3: All comments have been addressed

Reviewer #4: (No Response)

2. Is the manuscript technically sound, and do the data support the conclusions?

Reviewer #2: Yes

Reviewer #3: Yes

Reviewer #4: Partly

3. Has the statistical analysis been performed appropriately and rigorously? 

Reviewer #2: Yes

Reviewer #3: Yes

Reviewer #4: N/A

4. Have the authors made all data underlying the findings in their manuscript fully available?

Reviewer #2: Yes

Reviewer #3: Yes

Reviewer #4: Yes

5. Is the manuscript presented in an intelligible fashion and written in standard English?

Reviewer #2: Yes

Reviewer #3: Yes

Reviewer #4: Yes

6. Review Comments to the Author

Reviewer #2: Minor fixes.

-Numbering of sections is broken (2.2, 2.3, 2.3)

-The description of the hypothesis H2 is very short (compared to the description of the hypothesis H1)

Reviewer #3: All suggestions are carefully addressed and I am glad to recommend to publish this interesting paper.

Reviewer #4: Summary

This paper investigates the impact of Chinese listed companies R&D investments on investors’ short-term performance expectations, through adopting event study method to obtain the market reaction of performance forecasts. The authors find that corporate R&D investments help to reduce the investors’ short-term performance expectations and the investors show more tolerance to the corporates with high R&D investments. Moreover, institutional investors shareholdings have negative impact on the tolerance effect and the implementation of China’s innovation-driven strategy promotes the tolerance effect and thus corporate innovation.

Comments

1. The references with high quality cited in this paper are far from sufficient and supportive as present, especially for identifying the investors’ short-term performance expectations. In this paper, it is a key point to find an appropriate proxy variable for the investors’ expectations, which are usually discussed in theoretical model in the extant literature. It is required a convincing explanation with literature supporting for illustrating the validity and accuracy of the identification method in empirical analysis.

2. Combined with Chinese institutional and regulation background, generally, corporates with material performance change or considerable loss are required to release performance forecasts compulsively. That is to say, the sample companies are likely to have unstable cash flow and stock price, leading to the prominence of short-term holding speculation rather than long-term investment. Besides, Heckman model shows the existence of sample selection bias, which need to be addressed more prudentially and thoroughly.

3. As the control variables are limited relatively, firm fixed effect is suggested to adopt in regression to mitigate the endogeneity problem of omitting variables.

4. There is a lack of definition and introduction of the variable 〖URF〗_it in the regression.

Reference

Adcock, C., Hua, X., Mazouz, K., & Yin, S. (2014). Does the stock market reward innovation? European stock index reaction to negative news during the global financial crisis. Journal of International Money and Finance, 49(Part B), 470–491.

Chan, L. K. C., Lakonishok, J., & Sougiannis, T. (2001). The Stock Market Valuation of Research and Development Expenditures. Journal of Finance, 56(6), 2431–2456.

David, D. , & Paolo, F. . (2020). Uncertainty, investor sentiment, and innovation. The Review of Financial Studies(3), 3.

Howell, S. T. (2017). Financing Innovation: Evidence from R&D Grants. American Economic Review, 107(4), 1136–1164.

Oh, J.-M. (2017). Absorptive Capacity, Technology Spillovers, and the Cross-Section of Stock Returns. Journal of Banking and Finance, 85, 146–164.

Song, J., Su, Z., & Nie, X. (2018). Does development of financial markets help firm innovation? Evidence from China. Economic & Political Studies, 6(2), 194–208.

Wu, Q., Zheng, L., & Hasan, T. (2022). CEOs’ political ideologies and innovation: Evidence from US public firms. Economic & Political Studies, 10(3), 353–367.

7. PLOS authors have the option to publish the peer review history of their article (what does this mean?). If published, this will include your full peer review and any attached files.

Reviewer #2: No

Reviewer #3: No

Reviewer #4: No

---

## [Editor Report · Acceptance letter]

2 Jan 2023

PONE-D-22-20174R2 

Will R&D make investors more tolerant? Analysis based on the performance forecast of Chinese listed companies 

Dear Dr. Xu:

I'm pleased to inform you that your manuscript has been deemed suitable for publication in PLOS ONE. Congratulations! Your manuscript is now with our production department. 

Kind regards, 

on behalf of

Dr. Maurizio Fiaschetti 

Academic Editor

PLOS ONE